# Phase Transitions in the Pooled Data Problem

**Jonathan Scarlett and Volkan Cevher**

Laboratory for Information and Inference Systems (LIONS)
École Polytechnique Fédérale de Lausanne (EPFL)
{jonathan.scarlett,volkan.cevher}@epfl.ch

## Abstract

In this paper, we study the *pooled data* problem of identifying the labels associated with a large collection of items, based on a sequence of pooled tests revealing the counts of each label within the pool. In the noiseless setting, we identify an exact asymptotic threshold on the required number of tests with optimal decoding, and prove a *phase transition* between complete success and complete failure. In addition, we present a novel *noisy* variation of the problem, and provide an information-theoretic framework for characterizing the required number of tests for general random noise models. Our results reveal that noise can make the problem considerably more difficult, with strict increases in the scaling laws even at low noise levels. Finally, we demonstrate similar behavior in an *approximate recovery* setting, where a given number of errors is allowed in the decoded labels.

## 1 Introduction

Consider the following setting: There exists a large population of items, each of which has an associated label. The labels are initially unknown, and are to be estimated based on *pooled tests*. Each pool consists of some subset of the population, and the test outcome reveals the *total number of items* corresponding to each label that are present in the pool (but not the individual labels). This problem, which we refer to as the *pooled data* problem, was recently introduced in [1,2], and further studied in [3,4]. It is of interest in applications such as medical testing, genetics, and learning with privacy constraints, and has connections to the group testing problem [5] and its linear variants [6,7].

The best known bounds on the required number of tests under optimal decoding were given in [3]; however, the upper and lower bounds therein do not match, and can exhibit a large gap. In this paper, we completely close these gaps by providing a new lower bound that exactly matches the upper bound of [3]. These results collectively reveal a *phase transition* between success and failure, with the probability of error vanishing when the number of tests exceeds a given threshold, but tending to one below that threshold. In addition, we explore the novel aspect of random noise in the measurements, and show that this can significantly increase the required number of tests. Before summarizing these contributions in more detail, we formally introduce the problem.

### 1.1 Problem setup

We consider a large population of items $[p] = \{1, \ldots, p\}$, each of which has an associated label in $[d] = \{1, \ldots, d\}$. We let $\pi = (\pi_1, \ldots, \pi_d)$ denote a vector containing the proportions of items having each label, and we assume that the vector of labels itself, $\beta = (\beta_1, \ldots, \beta_p)$, is uniformly distributed over the sequences consistent with these proportions:

$$\beta \sim \text{Uniform}(\mathcal{B}(\pi)), \tag{1}$$

where $\mathcal{B}(\pi)$ is the set of length-$p$ sequences whose empirical distribution is $\pi$.

The goal is to recover $\beta$ based on a sequence of pooled tests. The $i$-th test is represented by a (possibly random) vector $X^{(i)} \in \{0, 1\}^p$, whose $j$-th entry $X_j^{(i)}$ indicates whether the $j$-th item is

| Sufficient for $P_{\mathrm{e}} \to 0$ [3] | Necessary for $P_{\mathrm{e}} \not\to 1$ [3] | Necessary for $P_{\mathrm{e}} \not\to 1$ (this paper) |
|---|---|---|
| $\dfrac{p}{\log p} \cdot \displaystyle\max_{r \in \{1,\ldots,d-1\}} f(r)$ | $\dfrac{p}{\log p} \cdot \dfrac{1}{2} f(1)$ | $\dfrac{p}{\log p} \cdot \displaystyle\max_{r \in \{1,\ldots,d-1\}} f(r)$ |

Table 1: Necessary and sufficient conditions on the number of tests $n$ in the noiseless setting. The function $f(r)$ is defined in (5). Asymptotic multiplicative $1 + o(1)$ terms are omitted.

| Noiseless testing | Noisy testing (SNR $= p^{\Theta(1)}$) | Noisy testing (SNR $= (\log p)^{\Theta(1)}$) | Noisy testing (SNR $= \Theta(1)$) |
|---|---|---|---|
| $\Theta\left(\dfrac{p}{\log p}\right)$ | $\Omega\left(\dfrac{p}{\log p}\right)$ | $\Omega\left(\dfrac{p}{\log \log p}\right)$ | $\Omega\big(p \log p\big)$ |

Table 2: Necessary and sufficient conditions on the number of tests $n$ in the noisy setting. SNR denotes the signal-to-noise ratio, and the noise model is given in Section 2.2.

included in the $i$-th test. We define a *measurement matrix* $\mathbf{X} \in \{0,1\}^{n \times p}$ whose $i$-th row is given by $X^{(i)}$ for $i = 1, \ldots, n$, where $n$ denotes the total number of tests. We focus on the *non-adaptive* testing scenario, where the entire matrix $\mathbf{X}$ must be specified prior to performing any tests.

In the noiseless setting, the $i$-th test outcome is a vector $Y^{(i)} = (Y_1^{(i)}, \ldots, Y_d^{(i)})$, with $t$-th entry

$$Y_t^{(i)} = N_t(\beta, X^{(i)}), \tag{2}$$

where for $t = 1, \ldots, d$, we let $N_t(\beta, X) = \sum_{j \in [p]} \mathbb{1}\{\beta_j = t \cap X_j = 1\}$ denote the number of items with label $t$ that are included in the test described by $X \in \{0,1\}^p$. More generally, in the possible presence of noise, the $i$-th observation is randomly generated according to

$$\big(Y^{(i)} \mid X^{(i)}, \beta\big) \sim P_{Y \mid N_1(\beta, X^{(i)}) \ldots N_d(\beta, X^{(i)})} \tag{3}$$

for some conditional probability mass function $P_{Y \mid N_1, \ldots, N_d}$ (or density function in the case of continuous observations). We assume that the observations $Y^{(i)}$ ($i = 1, \ldots, n$) are conditionally independent given $\mathbf{X}$, but otherwise make no assumptions on $P_{Y \mid N_1, \ldots, N_d}$. Clearly, the noiseless model (2) falls under this more general setup.

Similarly to $\mathbf{X}$, we let $\mathbf{Y}$ denote an $n \times d$ matrix of observations, with the $i$-th row being $Y^{(i)}$. Given $\mathbf{X}$ and $\mathbf{Y}$, a *decoder* outputs an estimate $\hat{\beta}$ of $\beta$, and the error probability is given by

$$P_{\mathrm{e}} = \mathbb{P}[\hat{\beta} \neq \beta], \tag{4}$$

where the probability is with respect to $\beta$, $\mathbf{X}$, and $\mathbf{Y}$. We seek to find conditions on the number of tests $n$ under which $P_{\mathrm{e}}$ attains a certain target value in the limit as $p \to \infty$, and our main results provide necessary conditions (i.e., lower bounds on $n$) for this to occur. As in [3], we focus on the case that $d$ and $\pi$ are fixed and do not depend on $p$.[1]

## 1.2 Contributions and comparisons to existing bounds

Our focus in this paper is on *information-theoretic* bounds on the required number of tests that hold regardless of practical considerations such as computation and storage. Among the existing works in the literature, the one most relevant to this paper is [3], whose bounds strictly improve on the initial bounds in [1]. The same authors also proved a phase transition for a *practical* algorithm based on approximate message passing [4], but the required number of tests is in fact significantly larger than the information-theoretic threshold (specifically, linear in $p$ instead of sub-linear).

Table 1 gives a summary of the bounds from [3] and our contributions in the noiseless setting. To define the function $f(r)$ therein, we introduce the additional notation that for $r = \{1, \ldots, d-1\}$, $\pi^{(r)} = (\pi_1^{(r)}, \ldots, \pi_r^{(r)})$ is a vector whose first entry sums the largest $d - r + 1$ entries of $\pi$, and whose remaining entries coincide with the remaining $r - 1$ entries of $\pi$. We have

$$f(r) = \max_{r \in \{1,\ldots,d-1\}} \frac{2(H(\pi) - H(\pi^{(r)}))}{d - r}, \tag{5}$$

meaning that the entries in Table 1 corresponding to the results of [3] are given as follows:

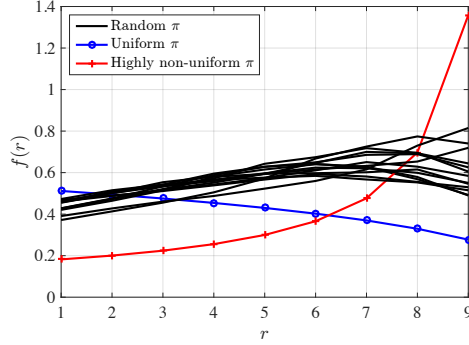

Figure 1: The function $f(r)$ in (5), for several choices of $\pi$, with $d = 10$. The random $\pi$ are drawn uniformly on the probability simplex, and the highly non-uniform choice of $\pi$ is given by $\pi = (0.49, 0.49, 0.0025, \ldots, 0.0025)$. When the maximum is achieved at $r = 1$, the bounds of [3] coincide up to a factor of two, whereas if the maximum is achieved for $r > 1$ then the gap is larger.

- (Achievability) When the entries of $\mathbf{X}$ are i.i.d. on Bernoulli$(q)$ for some $q \in (0,1)$ (not depending on $p$), there exists a decoder such that $P_e \to 0$ as $p \to \infty$ with

$$n \leq \frac{p}{\log p} \left( \max_{r \in \{1, \ldots, d-1\}} \frac{2(H(\pi) - H(\pi^{(r)}))}{d - r} \right)(1 + \eta) \tag{6}$$

  for arbitrarily small $\eta > 0$.

- (Converse) In order to achieve $P_e \nrightarrow 1$ as $p \to \infty$, it is necessary that

$$n \geq \frac{p}{\log p} \left( \frac{H(\pi)}{d - 1} \right)(1 - \eta) \tag{7}$$

  for arbitrarily small $\eta > 0$.

Unfortunately, these bounds do not coincide. If the maximum in (6) is achieved by $r = 1$ (which occurs, for example, when $\pi$ is uniform [3]), then the gap only amounts to a factor of two. However, as we show in Figure 1, if we compute the bounds for some "random" choices of $\pi$ then the gap is typically larger (i.e., $r = 1$ does not achieve the maximum), and we can construct choices where the gap is significantly larger. Closing these gaps was posed as a key open problem in [3].

We can now summarize our contributions as follows:

1. We give a lower bound that exactly matches (6), thus completely closing the above-mentioned gaps in the existing bounds and solving the open problem raised in [3]. More specifically, we show that $P_e \to 1$ whenever $n \leq \frac{p}{\log p} \left( \max_{r \in \{1, \ldots, d-1\}} \frac{2(H(\pi) - H(\pi^{(r)}))}{d - r} \right)(1 - \eta)$ for some $\eta > 0$, thus identifying an exact *phase transition* – a threshold above which the error probability vanishes, but below which the error probability tends to one.

2. We develop a framework for understanding variations of the problem consisting of random noise, and give an example of a noise model where the scaling laws are strictly higher compared to the noiseless case. A summary is given in Table 2; the case $\mathrm{SNR} = (\log p)^{\Theta(1)}$ reveals a strict increase in the scaling laws even when the signal-to-noise ratio grows unbounded, and the case $\mathrm{SNR} = \Theta(1)$ reveals that the required number of tests increases from sub-linear to super-linear in the dimension when the signal-to-noise ratio is constant.

3. In the supplementary material, we discuss how our lower bounds extend readily to the *approximate recovery* criterion, where we only require $\beta$ to be identified up to a certain Hamming distance. However, for clarity, we focus on exact recovery throughout the paper.

In a recent independent work [8], an *adversarial* noise setting was introduced. This turns out to be fundamentally different to our noisy setting. In particular, the results of [8] state that exact recovery is impossible, and even with approximate recovery, a huge number of tests (i.e., higher than polynomial) is needed unless $\Delta = O\big(q_{\max}^{1/2 + o(1)}\big)$, where $q_{\max}$ is the maximum allowed reconstruction error measured by the Hamming distance, and $\Delta$ is maximum adversarial noise amplitude. Of course, both random and adversarial noise are of significant interest, depending on the application.

**Notation.** For a positive integer $d$, we write $[d] = \{1, \ldots, d\}$. We use standard information-theoretic notations for the (conditional) entropy and mutual information, e.g., $H(X)$, $H(Y|X)$, $I(X;Y|Z)$ [9]. All logarithms have base $e$, and accordingly, all of the preceding information measures are in units of nats. The Gaussian distribution with mean $\mu$ and variance $\sigma^2$ is denoted by $\mathrm{N}(\mu, \sigma^2)$. We use the standard asymptotic notations $O(\cdot)$, $o(\cdot)$, $\Omega(\cdot)$, $\omega(\cdot)$ and $\Theta(\cdot)$.

## 2 Main results

In this section, we present our main results for the noiseless and noisy settings. The proofs are given in Section 3, as well as the supplementary material.

### 2.1 Phase transition in the noiseless setting

The following theorem proves that the upper bound given in (6) is tight. Recall that for $r = \{1, \ldots, d-1\}$, $\pi^{(r)} = (\pi_1^{(r)}, \ldots, \pi_r^{(r)})$ is a vector whose first entry sums the largest $d - r + 1$ entries of $\pi$, and whose remaining entries coincide with the remaining $r - 1$ entries of $\pi$.

**Theorem 1.** (Noiseless setting) *Consider the pooled data problem described in Section 1.1 with a given number of labels $d$ and label proportion vector $\pi$ (not depending on the dimension $p$). For any decoder, in order to achieve $P_{\mathrm{e}} \not\to 1$ as $p \to \infty$, it is necessary that*

$$n \geq \frac{p}{\log p} \left( \max_{r \in \{1, \ldots, d-1\}} \frac{2(H(\pi) - H(\pi^{(r)}))}{d - r} \right)(1 - \eta) \tag{8}$$

*for arbitrarily small $\eta > 0$.*

Combined with (6), this result reveals an exact *phase transition* on the required number of measurements: Denoting $n^* = \frac{p}{\log p}\left( \max_{r \in \{1, \ldots, d-1\}} \frac{2(H(\pi) - H(\pi^r))}{d - r} \right)$, the error probability vanishes for $n \geq n^*(1 + \eta)$, tends to one for $n \leq n^*(1 - \eta)$, regardless of how small $\eta$ is chosen to be.

**Remark 1.** Our model assumes that $\beta$ is uniformly distributed over the sequences with empirical distribution $\pi$, whereas [3] assumes that $\beta$ is i.i.d. on $\pi$. However, Theorem 1 readily extends to the latter setting: Under the i.i.d. model, once we condition on a given empirical distribution, the conditional distribution of $\beta$ is uniform. As a result, the converse bound for the i.i.d. model follows directly from Theorem 1 by basic concentration and the continuity of the entropy function.

### 2.2 Information-theoretic framework for the noisy setting

We now turn to *general noise models* of the form (3), and provide necessary conditions for the noisy pooled data problem in terms of the mutual information. General characterizations of this form were provided previously for group testing [10, 11] and other sparse recovery problems [12, 13].

Our general result is stated in terms of a maximization over a vector parameter $\boldsymbol{\ell} = (\ell_1, \ldots, \ell_d)$ with $\ell_t \in \{0, \ldots, \pi_t p\}$ for all $t$. We will see in the proof that $\ell_t$ represents the number of items of type $t$ that are unknown to the decoder after $p\pi_t - \ell_t$ are revealed by a genie. We define the following:

- Given $\boldsymbol{\ell}$ and $\beta$, we let $S_{\boldsymbol{\ell}}$ be a random set of indices in $[p]$ such that for each $t \in [d]$, the set contains $\ell_t$ indices corresponding to entries where $\beta$ equals $t$. Specifically, we define $S_{\boldsymbol{\ell}}$ to be uniformly distributed over all such sets. Moreover, we define $S_{\boldsymbol{\ell}}^c = [p] \setminus S_{\boldsymbol{\ell}}$.

- Given the above definitions, we define

$$\beta_{S_{\boldsymbol{\ell}}^c} = \begin{cases} \beta_j & j \in S_{\boldsymbol{\ell}}^c \\ \star & \text{otherwise,} \end{cases} \tag{9}$$

  where $\star$ can be thought of as representing an unknown value. Hence, knowing $\beta_{S_{\boldsymbol{\ell}}^c}$ amounts to knowing the labels of all items in the set $S_{\boldsymbol{\ell}}^c$.

- We define $|\mathcal{B}_{\boldsymbol{\ell}}(\pi)|$ to be the number of sequences $\beta \in [d]^p$ that coincide with a given $\beta_{S_{\boldsymbol{\ell}}^c}$ on the entries not equaling $\star$, while also having empirical distribution $\pi$ overall. This number does not depend on the specific choice of $S_{\boldsymbol{\ell}}^c$. As an example, when $\ell_t = p\pi_t$ for all $t$, we have $S_{\boldsymbol{\ell}} = [p]$, $\beta_{S_{\boldsymbol{\ell}}^c} = (\star, \ldots, \star)$, and $|\mathcal{B}_{\boldsymbol{\ell}}(\pi)| = |\mathcal{B}(\pi)|$, defined following (1)

- We let $\|\boldsymbol{\ell}\|_0$ denote the number of values in $(\ell_1, \ldots, \ell_d)$ that are positive.

With these definitions, we have the following result for general random noise models.

**Theorem 2.** (Noisy setting) *Consider the pooled data problem described in Section 1.1 under a general observation model of the form* (3)*, with a given number of labels d and label proportion vector $\pi$. For any decoder, in order to achieve $P_{\mathrm{e}} \leq \delta$ for a given $\delta \in (0, 1)$, it is necessary that*

$$n \geq \max_{\boldsymbol{\ell} \,:\, \|\boldsymbol{\ell}\|_0 \geq 2} \frac{\big(\log |\mathcal{B}_{\boldsymbol{\ell}}(\pi)|\big)(1 - \delta) - \log 2}{\frac{1}{n} \sum_{i=1}^{n} I(\beta; Y^{(i)} | \beta_{S_{\boldsymbol{\ell}}^c}, X^{(i)})}. \tag{10}$$

In order to obtain more explicit bounds on $n$ from (10), one needs to characterize the mutual information terms, ideally forming an upper bound that does not depend on the distribution of the measurement matrix $\mathbf{X}$. We do this for some specific models below; however, in general it can be a difficult task. The following corollary reveals that if the entries of $\mathbf{X}$ are i.i.d. on $\mathrm{Bernoulli}(q)$ for some $q \in (0, 1)$ (as was assumed in [3]), then we can simplify the bound.

**Corollary 1.** (Noisy setting with Bernoulli testing) *Suppose that the entries of $\mathbf{X}$ are i.i.d. on* $\mathrm{Bernoulli}(q)$ *for some* $q \in (0, 1)$*. Under the setup of Theorem 2, it is necessary that*

$$n \geq \max_{\boldsymbol{\ell} \,:\, \|\boldsymbol{\ell}\|_0 \geq 2} \frac{\big(\log |\mathcal{B}_{\boldsymbol{\ell}}(\pi)|\big)(1 - \delta) - \log 2}{I(X_{0,\boldsymbol{\ell}}; Y | X_{1,\boldsymbol{\ell}})}, \tag{11}$$

*where $(X_{0,\boldsymbol{\ell}}, X_{1,\boldsymbol{\ell}}, Y)$ are distributed as follows: (i) $X_{0,\boldsymbol{\ell}}$ (respectively, $X_{1,\boldsymbol{\ell}}$) is a concatenation of the vectors $X_{0,\boldsymbol{\ell}}(1), \ldots, X_{0,\boldsymbol{\ell}}(d)$ (respectively, $X_{1,\boldsymbol{\ell}}(1), \ldots, X_{1,\boldsymbol{\ell}}(d)$), the t-th of which contains $\ell_t$ (respectively, $\pi_t p - \ell_t$) entries independently drawn from $\mathrm{Bernoulli}(q)$; (ii) Letting each $N_t$ ($t = 1, \ldots, d$) be the total number of ones in $X_{0,\boldsymbol{\ell}}(t)$ and $X_{1,\boldsymbol{\ell}}(t)$ combined, the random variable $Y$ is drawn from $P_{Y|N_1,\ldots,N_d}$ according to* (3)*.*

As well as being simpler to evaluate, this corollary may be of interest in scenarios where one does not have complete freedom in designing $\mathbf{X}$, and one instead insists on using Bernoulli testing. For instance, one may not know how to optimize $\mathbf{X}$, and accordingly resort to generating it at random.

**Example 1: Application to the noiseless setting.** In the supplementary material, we show that in the noiseless setting, Theorem 2 recovers a weakened version of Theorem 1 with $1 - \eta$ replaced by $1 - \delta - o(1)$ in (8). Hence, while Theorem 2 does not establish a phase transition, it does recover the exact threshold on the number of measurements required to obtain $P_{\mathrm{e}} \to 0$.

An overview of the proof of this claim is as follows. We restrict the maximum in (10) to choices of $\boldsymbol{\ell}$ where each $\ell_t$ equals either its minimum value 0 or its maximum value $p\pi_t$. Since we are in the noiseless setting, each mutual information term reduces to the conditional entropy of $Y^{(i)} = (Y_1^{(i)}, \ldots, Y_d^{(i)})$ given $\beta_{S_{\boldsymbol{\ell}}^c}$ and $X^{(i)}$. For the values of $t$ such that $\ell_t = 0$, the value $Y_t^{(i)}$ is deterministic (i.e., it has zero entropy), whereas for the values of $t$ such that $\ell_t = p\pi_t$, the value $Y_t^{(i)}$ follows a hypergeometric distribution, whose entropy behaves as $\big(\frac{1}{2}\log p\big)(1 + o(1))$.

In the case that $\mathbf{X}$ is i.i.d. on $\mathrm{Bernoulli}(q)$, we can use Corollary 1 to obtain the following necessary condition for $P_{\mathrm{e}} \leq \delta$ as as $p \to \infty$, proved in the supplementary material:

$$n \geq \frac{p}{\log(pq(1 - q))} \bigg( \max_{r \in \{1, \ldots, d-1\}} \frac{2(H(\pi) - H(\pi^r))}{d - r} \bigg)(1 - \delta - o(1)) \tag{12}$$

for any $q = q(p)$ such that both $q$ and $1 - q$ behave as $\omega\big(\frac{1}{p}\big)$. Hence, while $q = \Theta(1)$ recovers the threshold in (8), the required number of tests strictly increases when $q = o(1)$, albeit with a mild logarithmic dependence.

**Example 2: Group testing.** To highlight the versatility of Theorem 2 and Corollary 1, we show that the latter recovers the lower bounds given in the group testing framework of [11].

Set $d = 2$, and let label 1 represent "defective" items, and label 2 represent "non-defective" items. Let $P_{Y|N_1 N_2}$ be of the form $P_{Y|N_1}$ with $Y \in \{0, 1\}$, meaning the observations are binary and depend only on the number of defective items in the test. For brevity, let $k = p\pi_1$ denote the total number of defective items, so that $p\pi_2 = p - k$ is the number of non-defective items.

Letting $\ell_2 = p - k$ in (11), and letting $\ell_1$ remain arbitrary, we obtain the necessary condition

$$n \geq \max_{\ell_1 \in \{1, \ldots, k\}} \frac{\big(\log \binom{p-k+\ell_1}{\ell_1}\big)(1 - \delta) - \log 2}{I(X_{0,\ell_1}; Y | X_{1,\ell_1})}, \tag{13}$$

where $X_{0,\ell_1}$ is a shorthand for $X_{0,\ell}$ with $\ell = (\ell_1, p - k)$, and similarly for $X_{1,\ell_1}$. This matches the lower bound given in [11] for Bernoulli testing with general noise models, for which several corollaries for specific models were also given.

**Example 3: Gaussian noise.** To give a concrete example of a noisy setting, consider the case that we observe the values in (2), but with each such value corrupted by independent Gaussian noise:
$$Y_t^{(i)} = N_t(\beta, X^{(i)}) + Z_t^{(i)}, \tag{14}$$
where $Z_t^{(i)} \sim \mathrm{N}(0, p\sigma^2)$ for some $\sigma^2 > 0$. Note that given $X^{(i)}$, the values $N_t$ themselves have variance at most proportional to $p$ (e.g., see Appendix C), so $\sigma^2 = \Theta(1)$ can be thought of as the constant signal-to-noise ratio (SNR) regime.

In the supplementary material, we prove the following bounds for this model:

- By letting each $\ell_t$ in (10) equal its minimum or maximum value analogously to the noiseless case above, we obtain the following necessary condition for $P_e \le \delta$ as $p \to \infty$:
$$n \ge \left( \max_{G \subseteq [d] : |G| \ge 2} \frac{p_G H(\pi_G)}{\sum_{t \in G} \frac{1}{2} \log \left(1 + \frac{\pi_t}{4\sigma^2}\right)} \right) (1 - \delta - o(1)), \tag{15}$$
  where $p_G := \sum_{t \in G} \pi_t p$, and $\pi_G$ has entries $\frac{\pi_t}{\sum_{t' \in G} \pi_{t'}}$ for $t \in G$. Hence, we have the following:
  - In the case that $\sigma^2 = p^{-c}$ for some $c \in (0, 1)$, each summand in the denominator simplifies to $\left(\frac{c}{2} \log p\right)(1 + o(1))$, and we deduce that compared to the noiseless case (cf., (8)), the asymptotic number of tests increases by at least a constant factor of $\frac{1}{c}$.
  - In the case that $\sigma^2 = (\log p)^{-c}$ for some $c > 0$, each summand in the denominator simplifies to $\left(\frac{c}{2} \log \log p\right)(1 + o(1))$, and we deduce that compared to the noiseless case, the asymptotic number of tests increases by at least a factor of $\frac{\log p}{c \log \log p}$. Hence, we observe a strict increase in the scaling laws despite the fact that the SNR grows unbounded.
  - While (15) also provides an $\Omega(p)$ lower bound for the case $\sigma^2 = \Theta(1)$, we can in fact do better via a different choice of $\ell$ (see below).
- By letting $\ell_1 = p\pi_1$, $\ell_2 = 1$, and $\ell_t = 0$ for $t = 3, \ldots, d$, we obtain the necessary condition
$$n \ge \left(4p\sigma^2 \log p\right)(1 - \delta - o(1)) \tag{16}$$
  for $P_e \le \delta$ as $p \to \infty$. Hence, if $\sigma^2 = \Theta(1)$, we require $n = \Omega(p \log p)$; this is super-linear in the dimension, in contrast with the sub-linear $\Theta\left(\frac{p}{\log p}\right)$ behavior observed in the noiseless case. Note that this choice of $\ell$ essentially captures the difficulty in identifying a *single* item, namely, the one corresponding to $\ell_2 = 1$.

These findings are summarized in Table 2; see also the supplementary material for extensions to the approximate recovery setting.

**Remark 2.** While it may seem unusual to add continuous noise to discrete observations, this still captures the essence of the noisy pooled data problem, and simplifies the evaluation of the mutual information terms in (10). Moreover, this converse bound immediately implies the same bound for the *discrete* model in which the noise consists of adding a Gaussian term, rounding, and clipping to $\{0, \ldots, p\}$, since the decoder could always choose to perform these operations as pre-processing.

## 3 Proofs

Here we provide the proof of Theorem 1, along with an overview of the proof of Theorem 2. The remaining proofs are given in the supplementary material.

### 3.1 Proof of Theorem 1

**Step 1: Counting typical outcomes.** We claim that it suffices to consider the case that $\mathbf{X}$ is deterministic and $\hat{\beta}$ is a deterministic function of $\mathbf{Y}$; to see this, we note that when either of these are random we have $P_e = \mathbb{E}_{\mathbf{X}, \hat{\beta}}[\mathbb{P}_\beta[\text{error}]]$, and the average is lower bounded by the minimum.

The following lemma, proved in the supplementary material, shows that for any $X^{(i)}$, each entry of the corresponding outcome $Y^{(i)}$ lies in an interval of length $O\left(\sqrt{p \log p}\right)$ with high probability.

**Lemma 1.** *For any deterministic test vector $X \in \{0,1\}^p$, and for $\beta$ uniformly distributed on $\mathcal{B}(\pi)$, we have for each $t \in [d]$ that*

$$\mathbb{P}\left[\left|N_t(\beta, X) - \mathbb{E}[N_t(\beta, X)]\right| > \sqrt{p \log p}\right] \leq \frac{2}{p^2}. \tag{17}$$

By Lemma 1 and the union bound, we have with probability at least $1 - \frac{2nd}{p^2}$ that $\left|N_t(\beta, X^{(i)}) - \mathbb{E}[N_t(\beta, X^{(i)})]\right| \leq \sqrt{p \log p}$ for all $i \in [n]$ and $t \in [d]$. Letting this event be denoted by $\mathcal{A}$, we have

$$P_{\mathrm{e}} \geq \mathbb{P}[\mathcal{A}] - \mathbb{P}[\mathcal{A} \cap \text{no error}] \geq 1 - \frac{2nd}{p^2} - \mathbb{P}[\mathcal{A} \cap \text{no error}]. \tag{18}$$

Next, letting $\mathbf{Y}(\beta) \in [p]^{n \times d}$ denote $\mathbf{Y}$ explicitly as a function of $\beta$ and similarly for $\hat{\beta}(\mathbf{Y}) \in [d]^p$, and letting $\mathcal{Y}_{\mathcal{A}}$ denote the set of matrices $\mathbf{Y}$ under which the event $\mathcal{A}$ occurs, we have

$$\mathbb{P}[\mathcal{A} \cap \text{no error}] = \frac{1}{|\mathcal{B}(\pi)|} \sum_{b \in \mathcal{B}(\pi)} \mathbb{1}\{\mathbf{Y}(b) \in \mathcal{Y}_{\mathcal{A}} \cap \hat{\beta}(\mathbf{Y}(b)) = b\} \tag{19}$$

$$\leq \frac{|\mathcal{Y}_{\mathcal{A}}|}{|\mathcal{B}(\pi)|}, \tag{20}$$

where (20) follows since each each $\mathbf{Y} \in \mathcal{Y}_{\mathcal{A}}$ can only be counted once in the summation of (19), due to the condition $\hat{\beta}(\mathbf{Y}(b)) = b$.

**Step 2: Bounding the set cardinalities.** By a standard combinatorial argument (e.g., [14, Ch. 2]) and the fact that $\pi$ is fixed as $p \to \infty$, we have

$$|\mathcal{B}(\pi)| = e^{p(H(\pi)+o(1))}. \tag{21}$$

To bound $|\mathcal{Y}_{\mathcal{A}}|$, first note that the entries of each $Y^{(i)} \in [p]^d$ sum to a deterministic value, namely, the number of ones in $X^{(i)}$. Hence, each $\mathbf{Y} \in \mathcal{Y}_{\mathcal{A}}$ is uniquely described by a sub-matrix of $\mathbf{Y} \in [p]^{n \times d}$ of size $n \times (d-1)$. Moreover, since $\mathcal{Y}_{\mathcal{A}}$ only includes matrices under which $\mathcal{A}$ occurs, each value in this sub-matrix only takes one of at most $2\sqrt{p \log p} + 1$ values. As a result, we have

$$|\mathcal{Y}_{\mathcal{A}}| \leq \left(2\sqrt{p \log p} + 1\right)^{n(d-1)}, \tag{22}$$

and combining (18)–(22) gives

$$P_{\mathrm{e}} \geq \frac{\left(2\sqrt{p \log p} + 1\right)^{n(d-1)}}{e^{p(H(\pi)+o(1))}} - \frac{2nd}{p^2}. \tag{23}$$

Since $d$ is constant, it immediately follows that $P_{\mathrm{e}} \to 1$ whenever $n \leq \frac{pH(\pi)}{(d-1)\log(2\sqrt{p \log p}+1)}(1 - \eta)$ for some $\eta > 0$. Applying $\log(2\sqrt{p \log p} + 1) = \left(\frac{1}{2}\log p\right)(1 + o(1))$, we obtain the following necessary condition for $P_{\mathrm{e}} \not\to 1$:

$$n \geq \frac{2pH(\pi)}{(d-1)\log p}(1 - \eta). \tag{24}$$

This yields the term in (8) corresponding to $r = 1$.

**Step 3: Genie argument.** Let $G$ be a subset of $[d]$ of cardinality at least two, and define $G^c = [d] \backslash G$. Moreover, define $\beta_{G^c}$ to be a length-$p$ vector with

$$(\beta_{G^c})_j = \begin{cases} \beta_j & \beta_j \in G^c \\ \star & \beta_j \in G, \end{cases} \tag{25}$$

where the symbol $\star$ can be thought of as representing an unknown value. We consider a modified setting in which a genie reveals $\beta_{G^c}$ to the decoder, i.e., the decoder knows the labels of all items for which the label lies in $G^c$, and is only left to estimate those in $G$. This additional knowledge can only make the pooled data problem easier, and hence, any lower bound in this modified setting remains valid in the original setting.

In the genie-aided setting, instead of receiving the full observation vector $Y^{(i)} = (Y_1^{(i)}, \ldots, Y_d^{(i)})$, it is equivalent to only be given $\{Y_j^{(i)} : j \in G\}$, since the values in $G^c$ are uniquely determined

from $\beta_{G^c}$ and $X^{(i)}$. This means that the genie-aided setting can be cast in the original setting with modified parameters: (i) $p$ is replaced by $p_G = \sum_{t \in G} \pi_t p$, the number of items with unknown labels; (ii) $d$ is replaced by $|G|$, the number of distinct remaining labels; (iii) $\pi$ is replaced by $\pi_G$, defined to be a $|G|$-dimensional probability vector with entries equaling $\frac{\pi_t}{\sum_{t' \in G} \pi_{t'}}$ $(t \in G)$.

Due to this equivalence, the condition (24) yields the necessary condition $n \geq \frac{2p_G H(\pi_G)}{(|G|-1)\log p}(1-\eta)$, and maximizing over all $G$ with $|G| \geq 2$ gives

$$n \geq \max_{G \subseteq [d]\,:\,|G| \geq 2} \frac{2p_G H(\pi_G)}{(|G|-1)\log p}\big(1-\eta\big). \tag{26}$$

**Step 4: Simplification.** Define $r = d - |G| + 1$. We restrict the maximum in (26) to sets $G$ indexing the highest $|G| = d - r + 1$ values of $\pi$, and consider the following process for sampling from $\pi$:

- Draw a sample $v$ from $\pi^{(r)}$ (defined above Theorem 1);
- If $v$ corresponds to the first entry of $\pi^{(r)}$, then draw a random sample from $\pi_G$ and output it as a label (i.e., the labels have conditional probability proportional to the top $|G|$ entries of $\pi$);
- Otherwise, if $v$ corresponds to one of the other entries of $\pi^{(r)}$, then output $v$ as a label.

By Shannon's property of entropy for sequentially-generated random variables [15, p. 10], we find that $H(\pi) = H(\pi^{(r)}) + \big(\sum_{t \in G} \pi_t\big) H(\pi_G)$. Moreover, since $p_G = p \cdot \sum_{t \in G} \pi_j$, this can be written as $p_G H(\pi_G) = p\big(H(\pi) - H(\pi^{(r)})\big)$. Substituting into (26), noting that $|G| - 1 = d - r$ by the definition of $r$, and maximizing over $r = 1, \ldots, d-1$, we obtain the desired result (8).

### 3.2 Overview of proof of Theorem 2

We can interpret the pooled data problem as a communication problem in which a "message" $\beta$ is sent over a "channel" $P_{Y|N_1,\ldots,N_d}$ via "codewords" of the form $\{(N_1^{(i)}, \ldots, N_d^{(i)})\}_{i=1}^n$ that are constructed by summing various columns of $\mathbf{X}$. As a result, it is natural to use Fano's inequality [9, Ch. 7] to lower bound the error probability in terms of information content (entropy) of $\beta$ and the amount of information that $\mathbf{Y}$ reveals about $\beta$ (mutual information).

However, a naive application of Fano's inequality only recovers the bound in (10) with $\ell = p\pi$. To handle the other possible choices of $\ell$, we again consider a *genie-aided setting* in which, for each $t \in [d]$, the decoder is informed of $p\pi_t - \ell_t$ of the items whose label equals $t$. Hence, it only remains to identify the remaining $\ell_t$ items of each type. This genie argument is a generalization of that used in the proof of Theorem 1, in which each $\ell_t$ was either equal to its minimum value zero or its maximum value $p\pi_t$. In Example 3 of Section 2, we saw that this generalization can lead to a strictly better lower bound in certain noisy scenarios.

The complete proof of Theorem 2 is given in the supplementary material.

## 4 Conclusion

We have provided novel information-theoretic lower bounds for the pooled data problem. In the noiseless setting, we provided a matching lower bound to the upper bound of [3], establishing an exact threshold indicating a phase transition between success and failure. In the noisy setting, we provided a characterization of general noise models in terms of the mutual information. In the special case of Gaussian noise, we proved an inherent added difficulty compared to the noiseless setting, with strict increases in the scaling laws even when the signal-to-noise ratio grows unbounded.

An interesting direction for future research is to provide *upper bounds* for the noisy setting, potentially establishing the tightness of Theorem 2 for general noise models. This appears to be challenging using existing techniques; for instance, the pooled data problem bears similarity to group testing with *linear* sparsity, whereas existing mutual information based upper bounds for group testing are limited to the *sub-linear* regime [10, 11, 16]. In particular, the proofs of such bounds are based on concentration inequalities which, when applied to the linear regime, lead to additional requirements on the number of tests that prevent tight performance characterizations.

**Acknowledgment:** This work was supported in part by the European Commission under Grant ERC Future Proof, SNF Sinergia project CRSII2-147633, SNF 200021-146750, and EPFL Fellows Horizon2020 grant 665667.

## Footnotes

[1]More precisely, $\pi$ should be rounded to the nearest empirical distribution (e.g., in $\ell_\infty$-norm) for sequences $\beta \in [d]^p$ of length $p$; we leave such rounding implicit throughout the paper.

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
