[Supplementary Material · PooledData_NIPS_supp.pdf]

# Supplementary Material

## "Phase Transitions in the Pooled Data Problem"

### (Jonathan Scarlett and Volkan Cevher, NIPS 2017)

Note that the references for the citations below are given in the main document.

## A   Proof of Lemma 1

Let $N_t$ be a shorthand for $N_t(\beta, X)$. Since $\beta$ uniformly distributed on the set of sequences with empirical distribution $\pi$, and $N_t$ counts the number of locations where $X_j = 1$ and $\beta_j = t$, we have $N_t \sim \mathrm{Hypergeometric}(\pi_t p, m(X), p)$, where $m(X)$ denotes the number of ones in $X$, and $\mathrm{Hypergeometric}(k, m, p)$ denotes the distribution of the number of "special items" when $k$ items are drawn from a population of $p$ items ($m$ of which are special) without replacement.

As a result, by Hoeffding's inequality for sampling without replacement [17], we have $\mathbb{P}[|N_1 - \mathbb{E}[N_1]| > \delta p] \le 2e^{-2\delta^2 p}$. Choosing $\delta = \sqrt{\frac{\log p}{p}}$ yields (17).

## B   Proofs of general results for the noisy setting

Throughout this section, the random variables $\mathbf{X}$ and $\beta$ are always discrete, whereas we allow the observations $\mathbf{Y}$ to be either discrete or continuous. In the continuous case, entropy terms should be interpreted as being the *differential entropy* [9, Ch. 8].

### B.1   Proof of Theorem 2

Throughout the proof, we make use of the definitions in Section 2.2 in terms of a given vector of integers $\boldsymbol{\ell} = (\ell_1, \ldots, \ell_d)$ with $0 \le \ell_t \le \pi_t p$. Note that we only consider choices of $\boldsymbol{\ell}$ such that $\|\boldsymbol{\ell}\|_0 \ge 2$, since otherwise the recovery problem would be trivial (e.g., if only a single $\ell_t$ is positive, then one achieves zero error probability be estimating all unknown labels to be $t$).

**Step 1: Fano's inequality and a genie argument.** As outlined in Section 3, a natural starting point is to apply *Fano's inequality* [9, Sec. 2.10] to obtain

$$I(\beta; \mathbf{Y}|\mathbf{X}) \ge \log|\mathcal{B}(\pi)| \cdot (1 - \delta) - \log 2. \tag{27}$$

Unfortunately, this bound alone is not sufficient to attain the desired result. To do that, we apply a *genie argument*, considering the following modified setting:

- The items $[p]$ are split into $S_{\boldsymbol{\ell}}$ (*cf.*, Section 2.2) and $S_{\boldsymbol{\ell}}^c = [p] \backslash S_{\boldsymbol{\ell}}$;
- A genie reveals to the decoder the labels of all items in $S_{\boldsymbol{\ell}}^c$, or equivalently, the vector $\beta_{S_{\boldsymbol{\ell}}^c}$ defined in (9);
- The decoder is left to identify only the entries in $\beta$ indexed by $S_{\boldsymbol{\ell}}$, i.e., to "fill in" the indices of $\beta_{S_{\boldsymbol{\ell}}^c}$ that are equal to $\star$.

Clearly the additional information at the decoder only makes the recovery problem easier, and thus any lower bound for the genie-aided setting is also a lower bound for the original setting.

Let us condition on particular realizations of $\beta_{S_{\boldsymbol{\ell}}^c} = b_{S_{\boldsymbol{\ell}}^c}$, and $\mathbf{X} = \mathbf{x}$, and let $\delta(b_{S_{\boldsymbol{\ell}}^c}, \mathbf{x})$ denote the corresponding conditional error probability. For any such realizations, the entries of $\beta$ indexed by $S_{\boldsymbol{\ell}}$ (i.e., locations where $b_{S_{\boldsymbol{\ell}}^c}$ equals $\star$) are uniform on the set of all possible subsequences that are consistent with $\pi$, of which there are $|\mathcal{B}_{\boldsymbol{\ell}}(\pi)|$ in total. Hence, Fano's inequality [9, Sec. 2.10] gives

$$I(\beta; \mathbf{Y}|\beta_{S_{\boldsymbol{\ell}}^c} = b_{S_{\boldsymbol{\ell}}^c}, \mathbf{X} = \mathbf{x}) \ge \log|\mathcal{B}_{\boldsymbol{\ell}}(\pi)| \cdot \left(1 - \delta(b_{S_{\boldsymbol{\ell}}^c}, \mathbf{x})\right) - \log 2, \tag{28}$$

and averaging both sides over $(\beta_{S_{\boldsymbol{\ell}}^c}, \mathbf{X})$ gives the following generalization of (27):

$$I(\beta; \mathbf{Y}|\beta_{S_{\boldsymbol{\ell}}^c}, \mathbf{X}) \ge \log|\mathcal{B}_{\boldsymbol{\ell}}(\pi)| \cdot (1 - \delta) - \log 2, \tag{29}$$

where we recall that $\delta$ is the target error probability. This provides the starting point of our analysis.

**Step 2: Bounding the mutual information.** We upper bound the conditional mutual information in (29) as

$$I(\beta; \mathbf{Y}|\beta_{S_{\boldsymbol{\ell}}^c}, \mathbf{X}) = H(\mathbf{Y}|\beta_{S_{\boldsymbol{\ell}}^c}, \mathbf{X}) - H(\mathbf{Y}|\beta, \beta_{S_{\boldsymbol{\ell}}^c}, \mathbf{X}) \tag{30}$$

$$= H(\mathbf{Y}|\beta_{S_{\boldsymbol{\ell}}^c}, \mathbf{X}) - \sum_{i=1}^n H(Y^{(i)}|\beta, \beta_{S_{\boldsymbol{\ell}}^c}, \mathbf{X}) \tag{31}$$

$$= H(\mathbf{Y}|\beta_{S_{\boldsymbol{\ell}}^c}, \mathbf{X}) - \sum_{i=1}^n H(Y^{(i)}|\beta, \beta_{S_{\boldsymbol{\ell}}^c}, X^{(i)}) \tag{32}$$

$$\leq \sum_{i=1}^n H(Y^{(i)}|\beta_{S_{\boldsymbol{\ell}}^c}, X^{(i)}) - \sum_{i=1}^n H(Y^{(i)}|\beta, \beta_{S_{\boldsymbol{\ell}}^c}, X^{(i)}) \tag{33}$$

$$= \sum_{i=1}^n I(\beta; Y^{(i)}|\beta_{S_{\boldsymbol{\ell}}^c}, X^{(i)}), \tag{34}$$

where:

- (31) follows since we have assumed that the observations are conditionally independent;
- (32) follows since given $(\beta, \beta_{S_{\boldsymbol{\ell}}^c})$, $Y^{(i)}$ depends on $\mathbf{X}$ only through $X^{(i)}$;
- (33) follows from the sub-additivity of entropy and the fact that conditioning reduces entropy (e.g., see [9, Ch. 2]).

Substituting (34) into (29), re-arranging, and maximizing over $\boldsymbol{\ell}$ (which was arbitrary in the above analysis), we obtain Theorem 2.

## B.2 Proof of Corollary 1

In the case that the entries of $\mathbf{X}$ are i.i.d. Bernoulli, each mutual information term $I(\beta; Y^{(i)}|\beta_{S_{\boldsymbol{\ell}}^c}, X^{(i)})$ is identical, and (10) becomes

$$n \geq \max_{\boldsymbol{\ell} \,:\, \|\boldsymbol{\ell}\|_0 \geq 2} \frac{\big(\log |\mathcal{B}_{\boldsymbol{\ell}}(\pi)|\big)(1-\delta) - \log 2}{I(\beta; Y|\beta_{S_{\boldsymbol{\ell}}^c}, X)}, \tag{35}$$

where we define $(X, Y) = (X^{(i)}, Y^{(i)})$ for some arbitrary fixed $i \in \{1, \dots, n\}$.

Let $X_{0,\boldsymbol{\ell}}$ (respectively, $X_{1,\boldsymbol{\ell}}$) be formed from $X$ by taking the sub-vector of $X$ indexed by $S_{\boldsymbol{\ell}}$ (respectively, $S_{\boldsymbol{\ell}}^c$) and re-ordering it so that the indices corresponding to class 1 appear first, then class 2, and so on. Since the entries of $X$ are i.i.d. Bernoulli, this means that the triplet $(X_{0,\boldsymbol{\ell}}, X_{1,\boldsymbol{\ell}}, Y)$ follows the joint distribution described in Theorem 2. We have

$$I(\beta; Y|\beta_{S_{\boldsymbol{\ell}}^c}, X) = H(Y|\beta_{S_{\boldsymbol{\ell}}^c}, X) - H(Y|\beta, \beta_{S_{\boldsymbol{\ell}}^c}, X) \tag{36}$$

$$= H(Y|X_{1,\boldsymbol{\ell}}, \beta_{S_{\boldsymbol{\ell}}^c}, X) - H(Y|X_{0,\boldsymbol{\ell}}, X_{1,\boldsymbol{\ell}}, \beta, \beta_{S_{\boldsymbol{\ell}}^c}, X) \tag{37}$$

$$\leq H(Y|X_{1,\boldsymbol{\ell}}) - H(Y|X_{0,\boldsymbol{\ell}}, X_{1,\boldsymbol{\ell}}, \beta, \beta_{S_{\boldsymbol{\ell}}^c}, X) \tag{38}$$

$$= H(Y|X_{1,\boldsymbol{\ell}}) - H(Y|X_{0,\boldsymbol{\ell}}, X_{1,\boldsymbol{\ell}}) \tag{39}$$

$$= I(X_{0,\boldsymbol{\ell}}; Y|X_{1,\boldsymbol{\ell}}), \tag{40}$$

where:

- (37) follows since $X_{1,\boldsymbol{\ell}}$ is a function of $(\beta_{S_{\boldsymbol{\ell}}^c}, X)$ and $(X_{0,\boldsymbol{\ell}}, X_{1,\boldsymbol{\ell}})$ is a function of $(\beta, \beta_{S_{\boldsymbol{\ell}}^c}, X)$;
- (38) follows since conditioning reduces entropy;
- (39) follows since $Y$ and $(\beta, \beta_{S_{\boldsymbol{\ell}}^c}, X)$ are conditionally independent given $(X_{0,\boldsymbol{\ell}}, X_{1,\boldsymbol{\ell}})$. This is because the model is of the form (3), and the values $\{N_t\}_{t=1}^d$ are already determined by $(X_{0,\boldsymbol{\ell}}, X_{1,\boldsymbol{\ell}})$.

Substituting (40) into (35) completes the proof.

## C  Applications of Theorem 2 to specific models

### C.1  Noiseless setting with arbitrary testing

Here we prove the first claim given in the application to the noiseless model following Theorem 2.

In the noiseless setting, $Y^{(i)}$ is a deterministic function of $(\beta, X^{(i)})$, and hence $I(\beta; Y^{(i)}|\beta_{S_\ell^c}, X^{(i)}) = H(Y^{(i)}|\beta_{S_\ell^c}, X^{(i)})$. It turns out to suffice to let $\ell = (\ell_1, \ldots, \ell_d)$ be such that each $\ell_t$ either equals its minimum value zero or its maximum value $p\pi_t$. We let $G \subseteq [d]$ index those equaling the maximum value, and let $G^c = [d]\backslash G$ index those equaling zero. As a result, $\beta_{S_\ell^c}$ in (9) is precisely equal to $\beta_{G^c}$ in (25), and we are left to bound $H(Y^{(i)}|\beta_{G^c}, X^{(i)})$. For notational simplicity, we focus on an arbitrary fixed value of $i$ and omit the superscripts $(\cdot)^{(i)}$.

Recall that $Y = (Y_1, \ldots, Y_d)$ according to (2). Given $G$, we let $G'$ be an arbitrary subset of $G$ with a single element removed, and we write $Y_G = (Y_t)_{t \in G}$, and similarly for $Y_{G^c}$ and $Y_{G'}$. With these definitions, we have

$$H(Y|\beta_{G^c}, X) = H(Y_G, Y_{G^c}|\beta_{G^c}, X) \tag{41}$$

$$= H(Y_{G'}, Y_{G^c}|\beta_{G^c}, X) \tag{42}$$

$$= H(Y_{G'}|\beta_{G^c}, X) \tag{43}$$

$$\leq \sum_{t \in G'} H(Y_t|\beta_{G^c}, X), \tag{44}$$

where (42) follows since any single entry of $Y$ can be uniquely determined as equaling the number of ones in $X$ minus the other $d-1$ entries, (43) follows since $Y_{G^c}$ is deterministic given $(\beta_{G^c}, X)$, and (44) follows from the sub-additivity of entropy.

We proceed by characterizing the conditional distribution of $Y_t$ for given values of $(\beta_{G^c}, X)$. Let $m_G$ denote the total number of ones in $X$ among the indices where $\beta_{G^c}$ equals $\star$ (i.e., the indices of items whose labels are in $G$). We denote these indices by $S_G$. Moreover, recall that $\beta$ is uniform on $\mathcal{B}(\pi)$, so once $\beta_{G^c}$ is known, the remaining entries are uniform on the set of possible outcomes consistent with both $\pi$ and $\beta_{S_\ell^c}$.

From these definitions and observations, we see that the items within $S_G$ having label $t$ are obtained by randomly selecting $\pi_t p$ indices uniformly at random without replacement from a total of $p_G := \sum_{t \in G} \pi_t p$ indices. Since $Y_t$ represents the number of such locations where $X$ equals one, we have

$$(Y_t|\beta_{G^c}, X) \sim \text{Hypergeometric}(\pi_t p, m_G, p_G). \tag{45}$$

Note that for $G = [d]$, this matches the distribution derived in Appendix A. Before proceeding, we present the following lemma regarding the entropy of an integer-valued random variable.

**Lemma 2.**  [18] *For any integer-valued random variable $U$, we have*

$$H(U) \leq \frac{1}{2}\log\left(2\pi e\left(\text{Var}[U] + \frac{1}{12}\right)\right). \tag{46}$$

Note that the right-hand side of (46) is the *differential entropy* of a Gaussian random variable with variance $\text{Var}[U] + \frac{1}{12}$ [9, Ch. 8]. For continuous random variables, an analogous result holds true without the addition of $\frac{1}{12}$, i.e., the Gaussian distribution maximizes entropy for a given variance.

For $U \sim \text{Hypergeometric}(k, m, p)$, we have

$$\text{Var}[U] = k \cdot \frac{m}{p} \cdot \frac{p-m}{p} \cdot \frac{p-k}{p-1} \leq \frac{k}{4}, \tag{47}$$

where we have applied $\frac{p-k}{p-1} \leq 1$ and $m(p-m) \leq \frac{p^2}{4}$. Hence, under the distribution in (45), the conditional variance of $Y_t$ is upper bounded by $\pi_t p/4$, and Lemma 2 yields

$$H(Y_t|\beta_{G^c}, X) \leq \frac{1}{2}\log\left(2\pi e\left(\frac{\pi_t p}{4} + \frac{1}{12}\right)\right) \tag{48}$$

$$= \left(\frac{1}{2}\log p\right)(1 + o(1)), \tag{49}$$

where in (49) we used the fact that $\pi$ does not depend on $p$ (and hence $\pi_t = \Theta(1)$ for all $t$). Substituting (49) into (44) and noting that $|G'| = |G|-1$, we obtain $H(Y|\beta_{G^c}, X) \leq \left(\frac{|G|-1}{2}\log p\right)(1+o(1))$.

Putting it all together, we have shown that $I(\beta; Y^{(i)}|\beta_{G^c}, X^{(i)}) \leq \left(\frac{|G|-1}{2}\log p\right)(1 + o(1))$ for all $i = 1, \ldots, n$. In addition, we have analogously to (21) that $\log|\mathcal{B}_\ell(\pi)| = p_G(H(\pi_G) + o(1))$ under our choice of $\ell$ (depending on $G$). Hence, substituting into (10), maximizing over $G$, and changing variables from $|G|$ to $r$ analogously to Section 3.1, we obtain (8) with $1 - \delta - o(1)$ in place of $1 - \eta$.

## C.2 Noiseless setting with Bernoulli testing

Here we derive (11) for the noiseless model with Bernoulli testing. We follow the same arguments as those used in Section C.1 for general tests, and therefore only describe the differences. We restrict the choices of $\ell$ as in Section C.1, indexing them by $G \subseteq [d]$ and using the definition of $\beta_{G^c}$ in (25). Moreover, we write $(X_G, X_{G^c})$ in place of $(X_{0,\ell}, X_{1,\ell})$.

The mutual information term $I(X_G; Y|X_{G^c})$ simplifies to $H(Y|X_{G^c})$ in the noiseless setting, and analogously to (44), we have

$$H(Y|X_{G^c}) = \sum_{t \in G'} H(Y_t|X_{G^c}), \tag{50}$$

where $G'$ is an arbitrary subset of $G$ with a single element removed. Next, we observe that each $Y_t$ for $t \in G'$ is in fact independent of $X_{G^c}$, and is distributed as $\mathrm{Binomial}(p\pi_t, q)$. The corresponding variance is $p\pi_t q(1 - q)$, and applying Lemma 2, we conclude that the entropy is upper bounded by $\frac{1}{2}\log\left(2\pi e\left(p\pi_t q(1-q) + \frac{1}{12}\right)\right)$. Since we have assumed that $pq$ and $p(1-q)$ both grow unbounded as $p \to \infty$, and recalling that $\pi_t = \Theta(1)$, this simplifies to $\left(\frac{1}{2}\log(pq(1-q))\right)(1 + o(1))$.

Once this upper bound on the entropy of each $Y_t$ is established, we deduce (12) using (11) and the same argument as that following (49).

## C.3 Gaussian noise with large signal-to-noise ratio

Here we derive the first bound (15) for the Gaussian noise model.

We again restrict the choices of $\ell$ as in Section C.1, indexing them by $G \subseteq [d]$ and using the definition of $\beta_{G^c}$ in (25). Letting $H(\cdot)$ denote the differential entropy [9, Ch. 8] of a continuous random variable, we have

$$I(\beta; Y|\beta_{G^c}, X) = H(Y|\beta_{G^c}, X) - H(Y|\beta, \beta_{G^c}, X) \tag{51}$$

$$= H(Y|\beta_{G^c}, X) - \frac{d}{2}\log(2\pi e p\sigma^2) \tag{52}$$

$$\leq \sum_{t=1}^{d} H(Y_t|\beta_{G^c}, X) - d\log(2\pi e p\sigma^2), \tag{53}$$

$$= \sum_{t \in G} H(Y_t|\beta_{G^c}, X) - (d - |G^c|)\log(2\pi e p\sigma^2), \tag{54}$$

where

- (52) follows since the only uncertainty in $Y$ given $(\beta, \beta_{G^c}, X)$ is that of the $d$ additive $\mathrm{N}(0, p\sigma^2)$ terms, each of which has differential entropy $\frac{1}{2}\log(2\pi e p\sigma^2)$ [9, Ch. 8];
- (53) follows from the sub-additivity of entropy;
- (54) follows since for $t \in G^c$, the only uncertainty in $Y_t$ given $(\beta_{G^c}, X)$ is that of the additive $\mathrm{N}(0, p\sigma^2)$ noise term.

For $t \in G$, each $Y_t$ is of the form $N_t + Z_t$, where $N_t$ is (conditionally) distributed as in (45), and $Z_t \sim \mathrm{N}(0, p\sigma^2)$ is independent of $N_t$. Using (47), we deduce that $\mathrm{Var}[Y_t|\beta_{G^c}, X] \leq p\pi_t/4 + p\sigma^2$ for any realizations of $(\beta_{G^c}, X)$, which in turn implies $H(Y_t|\beta_{G^c}, X) \leq \frac{1}{2}\log\left(2\pi e(p\pi_t/4 + p\sigma^2)\right)$ since the Gaussian distribution maximizes the differential entropy for a given variance [9, Thm. 8.6.5].

Substituting into (54) and noting that $d - |G^c| = |G|$, we obtain

$$I(\beta; Y | \beta_G, X) \le \sum_{t \in G} \frac{1}{2} \log \left( 2\pi e (p\pi_t/4 + p\sigma^2) \right) - |G| \log(2\pi e p\sigma^2) \tag{55}$$

$$= \sum_{t \in G} \frac{1}{2} \log \left( 1 + \frac{\pi_t}{4\sigma^2} \right). \tag{56}$$

In addition, as we already stated in the noiseless case, it holds that $\log |\mathcal{B}_{\boldsymbol{\ell}}(\pi)| = p_G(H(\pi_G) + o(1))$ under our choice of $\boldsymbol{\ell}$ (depending on $G$). Substituting into (10) and maximizing over $G$, we obtain the desired bound in (15).

## C.4 Gaussian noise with constant signal-to-noise ratio

Here we derive the second bound (16) for the Gaussian model.

We choose $\boldsymbol{\ell}$ in (8) with $\ell_1 = p\pi_1$, $\ell_2 = 1$, and $\ell_t = 0$ for $t = 3, \ldots, d$. Since only $p\pi_1 + 1$ entries of $\beta$ remain unspecified (i.e., the corresponding entries of $\beta_{S_{\boldsymbol{\ell}}^c}$ are equal to $\star$), and those become fully specified once we assign the single remaining item with label 2 (since this means the rest must have label 1), we have

$$|\mathcal{B}_{\boldsymbol{\ell}}(\pi)| = p\pi_1 + 1. \tag{57}$$

The main step is to bound the mutual information terms appearing in (8). We again focus on a single test indexed by $i$, and write $(X, Y)$ in place of $(X^{(i)}, Y^{(i)})$. We have

$$I(\beta; Y | \beta_{S_{\boldsymbol{\ell}}^c}, X) = I(\beta; Y_1, Y_2 | \beta_{S_{\boldsymbol{\ell}}^c}, X) \tag{58}$$

$$= H(Y_1, Y_2 | \beta_{S_{\boldsymbol{\ell}}^c}, X) - H(Y_1, Y_2 | \beta, \beta_{S_{\boldsymbol{\ell}}^c}, X) \tag{59}$$

$$= H(Y_1, Y_2 | \beta_{S_{\boldsymbol{\ell}}^c}, X) - \log(2\pi e(p\sigma^2)) \tag{60}$$

$$\le H(Y_1 | \beta_{S_{\boldsymbol{\ell}}^c}, X) + H(Y_2 | \beta_{S_{\boldsymbol{\ell}}^c}, X) - \log(2\pi e(p\sigma^2)), \tag{61}$$

where

- (58) follows since $Y_3, \ldots, Y_d$ are conditionally independent of $\beta$ given $(\beta_{S_{\boldsymbol{\ell}}^c}, X)$ (specifically, they are pure Gaussian noise due to the choice of $\boldsymbol{\ell}$);
- (60) follows since $Y_1$ and $Y_2$ are also pure Gaussian noise given $(\beta, \beta_{S_{\boldsymbol{\ell}}^c}, X)$, so they each have entropy $\frac{1}{2} \log(2\pi e(p\sigma^2))$;
- (61) follows from the sub-additivity of entropy.

To bound $H(Y_1 | \beta_{S_{\boldsymbol{\ell}}^c}, X)$, we recall that $Y_1 = N_1 + Z_1$, where $N_1$ counts the number of tested items with label 1, and $Z_1 \sim \mathrm{N}(0, p\sigma^2)$. We write this as $Y_1 = N_{\text{total}} - \xi + Z_1$, where $N_{\text{total}}$ is the total number of unspecified items included in the test (i.e., the number of $j \in [p]$ such that $(\beta_{S_{\boldsymbol{\ell}}^c})_j = \star$ and $X_j = 1$), and $\xi \in \{0, 1\}$ indicates whether the single unspecified item with label 2 is tested.

Since the quantity $N_{\text{total}}$ is deterministic given $(\beta_{S_{\boldsymbol{\ell}}^c}, X)$, the conditional variance of $Y_1$ is simply

$$\mathrm{Var}[Y_1 | \beta_{S_{\boldsymbol{\ell}}^c}, X] = \mathrm{Var}[-\xi + Z_1] \tag{62}$$

$$= \mathrm{Var}[\xi] + \mathrm{Var}[Z_1] \tag{63}$$

$$\le \frac{1}{4} + p\sigma^2, \tag{64}$$

where (63) follows since $\xi$ and $Z_1$ are independent, and (64) follows since a random variable on $\{0, 1\}$ has variance at most $\frac{1}{4}$, and since $Z_1$ is Gaussian with variance $p\sigma^2$. Finally, since the Gaussian distribution maximizes entropy for a given variance, we deduce that

$$H(Y_1 | \beta_{S_{\boldsymbol{\ell}}^c}, X) \le \frac{1}{2} \log \left( 2\pi e \left( \frac{1}{4} + p\sigma^2 \right) \right). \tag{65}$$

For $Y_2$, we apply the same argument, noting that $Y_2 = N_{2,\text{other}} + \xi + Z_2$, where $N_{2,\text{other}}$ counts the number of indices where $(\beta_{S_{\boldsymbol{\ell}}^c})_j = 2$ and $X_j = 1$. We see that $N_{2,\text{other}}$ is deterministic given

$(\beta_{S^c_\ell}, X)$, and it follows that $Y_2$ satisfies the same conditional variance bound as $Y_1$, and hence the same conditional entropy bound as (65).

Substituting (65) (and the analog for $Y_2$) into (61), we obtain

$$I(\beta; Y | \beta_{S^c_\ell}, X) \le \log(2\pi e(1/4 + p\sigma^2)) - \log(2\pi e(p\sigma^2)) \tag{66}$$

$$= \log\left(1 + \frac{1}{4p\sigma^2}\right) \tag{67}$$

$$\le \frac{1}{4p\sigma^2}, \tag{68}$$

where (68) follows from the inequality $\log(1 + \alpha) \le \alpha$. Finally, substituting (57) and (68) into (10) and writing $\log(p\pi_1 + 1) = (\log p)(1 + o(1))$, we obtain the desired result (16).

## D   Extensions to approximate recovery

Throughout the paper, we have considered the exact recovery criterion in (4), in which one insists on estimating every entry of $\beta$ correctly. However, both Theorems 1 and 2 extend readily to the *approximate recovery* setting, as we describe below. We note that relaxed recovery criteria are known to considerably reduce the number of measurements in certain problems such as compressive sensing [19, 20], while having a smaller effect in other problems including group testing [16, 21].

Suppose that we only require the recovery of $\beta$ up to a Hamming distance of $q_{\max} \in \{0, \dots, p\}$. Then the error probability is given by

$$P_e(q_{\max}) = \mathbb{P}\left[\sum_{j=1}^{p} \mathbb{1}\{\hat{\beta}_j \ne \beta_j\} > q_{\max}\right]. \tag{69}$$

One should certainly expect this criterion to reduce the number of measurements for certain values of $q_{\max}$: If $d = 2$ and $q_{\max} \ge \max\{p\pi_1, p\pi_2\}$ then we can achieve $P_e(q_{\max}) = 0$ with no tests, by simply declaring each entry of $\hat{\beta}$ to equal the most common label.

Nevertheless, the following generalization of Theorem 1 reveals that in the noiseless setting, the asymptotic reduction in the number of tests is insignificant when $q_{\max}$ is not too large.

**Theorem 3.** (Approximate recovery, noiseless) *Consider the noiseless pooled data problem under the approximate recovery criterion* (69), *with a given number of labels $d$ and label proportion vector $\pi$ (not depending on the dimension $p$), and a given maximum Hamming distance $q_{\max}$. Then for any decoder, in order to achieve $P_e(q_{\max}) \nrightarrow 1$ as $p \to \infty$, it is necessary that*

$$n \ge \frac{1}{\log p}\left(\max_{r \in \{1, \dots, d-1\}} \frac{2\big(pH(\pi) - pH(\pi^r) - \log \sum_{j=0}^{q_{\max}} \binom{p}{j}(d-1)^j\big)}{d - r}\right)(1 - \eta) \tag{70}$$

*for arbitrarily small $\eta > 0$.*

*Proof.* The proof is identical to that of Theorem 1 up until (19), at which point the condition $\hat{\beta}(\mathbf{Y}(b)) = b$ should be replaced by $d_H(\hat{\beta}(\mathbf{Y}(b)), b) \le q_{\max}$, where $d_H$ denotes the Hamming distance. The number of sequences within a Hamming distance $q_{\max}$ of a given $b \in [d]^p$ is upper bounded by $\sum_{j=0}^{q_{\max}} \binom{p}{j}(d-1)^j$, which follows by counting the number of ways of choosing $j \le q_{\max}$ locations and assigning one of $d - 1$ new values to each.

As a result, the right-hand side of (20) needs to be multiplied by $\sum_{j=0}^{\alpha^* p} \binom{p}{j}(d-1)^j$, and following the remainder of the proof with this factor incorporated, we obtain (70).   □

For any $q_{\max} = o(p)$, the term $\log \sum_{j=0}^{q_{\max}} \binom{p}{j}(d-1)^j$ is dominated by $pH(\pi) - pH(\pi^r)$, and hence the approximate recovery threshold is identical to the exact recovery threshold. Hence, a key implication of Theorem 3 is that asymptotically, recovering all labels is essentially as easy as recovering all but a vanishing fraction of the labels.

In contrast, if $q_{\max} = \alpha^* p$ for fixed $\alpha^* \in (0,1)$, the term $\log \sum_{j=0}^{q_{\max}} \binom{p}{j}(d-1)^j$ behaves as $\Theta(p)$, and Theorem 3 indicates that the approximate recovery criterion may permit improved constant factors in the required number of tests. However, the scaling laws are unchanged when $\alpha^*$ is sufficiently small (in particular, small enough to avoid the above-mentioned trivial cases).

Theorem 2 also extends naturally to the approximate recovery criterion, yielding the following.

**Theorem 4.** (Approximate recovery, noisy) *Consider the pooled data problem under a general observation model of the form* (3) *and the approximate recovery criterion* (69), *with a given number of labels $d$, label proportion vector $\pi$, and maximum Hamming distance $q_{\max}$. Then for any decoder, in order to achieve $P_{\mathrm{e}}(q_{\max}) \leq \delta$ for a given $\delta \in (0,1)$, it is necessary that*

$$n \geq \max_{\boldsymbol{\ell} \,:\, \|\boldsymbol{\ell}\|_0 \geq 2} \frac{\left( \log |\mathcal{B}_{\boldsymbol{\ell}}(\pi)| - \log \sum_{j=0}^{q_{\max}} \binom{p}{j}(d-1)^j \right)(1-\delta) - \log 2}{\frac{1}{n} \sum_{i=1}^{n} I(\beta; Y^{(i)} | \beta_{S_{\boldsymbol{\ell}}^c}, X^{(i)})}. \tag{71}$$

*Proof.* The proof is nearly identical to that of Theorem 2, except that we replace Fano's inequality by its counterpart for approximate recovery, analogously to previous works on problems such as support recovery [20, Appendix A] and graphical model selection [22, Lemma 1] (see also [23]). Similarly to the proof of Theorem 3, the term $\log \sum_{j=0}^{q_{\max}} \binom{p}{j}(d-1)^j$ represents the number of different $\hat{\beta}$ that remain feasible given that $\beta$ is fixed and an error does not occur. $\qquad \square$

An approximate recovery analog of Corollary 1 follows naturally from Theorem 4, as do bounds of the form (12)–(15) with analogous modifications to those given in (70).

On the other hand, Theorem 4 does not recover any meaningful analog of (16). This is because the proof of (16) is based on a choice of $\boldsymbol{\ell}$ with $\log |\mathcal{B}_{\boldsymbol{\ell}}(\pi)| \leq \log p$, which is dominated by $\log \sum_{j=0}^{q_{\max}} \binom{p}{j}(d-1)^j$ in (71) unless $q_{\max} = 0$. Stated differently, the proof of (16) essentially involves leaving the decoder with the difficulty of estimating one specific label, which is trivial in the approximate recovery setting.

Nevertheless, in the constant signal-to-noise ratio regime with either $q_{\max} = o(p)$ or $q_{\max} = \alpha^* p$ for sufficiently small $\alpha^* \in (0,1)$, one can still use the analog of (15) to prove an $\Omega(p)$ lower bound. While this is not as strong as the $\Omega(p \log p)$ bound proved for exact recovery, it still shows that noise increases the number of tests from sub-linear in the dimension to at least linear.