[Reviews · NeurIPS 2017]

Reviewer 1



Summary. This submission presents an information-theoretic study of the pooled data problem. The contributions are to provide a lower bound of the number of tests necessary for achieving decoding error less than 1 in the noiseless setting (Theorem 1), and a lower bound in noisy setting (Theorem 2, Corollary 1). The lower bound in the noiseless setting matches the previously obtained upper bound in the leading order in the number p of items, which in turn specifies the phase transition point between decodable and undecodable phases. Quality. The key ideas for proving Theorem 1, especially the genie argument that is used in reducing the problem to a simpler one with a smaller number of labels, are properly described, although I have not checked details of the proof. Clarity. In some places results are displayed without mentioning that they are asymptotic, i.e., in the limit p \to \infty, which might cause confusion. Originality. Showing existence of the phase transition in the noiseless setting is new. Studying the noisy version of the pooled data problem is another novel contribution. Significance. This work is of significance in that it has revealed existence of phase transition in the noiseless pooled data problem, as well as sample complexity lower bounds in the noisy setting. The latter would also encourage further studies on conditions under which perfect decoding should be possible in the noisy setting, as well as on efficient decoding algorithms from noisy observations. Minor points: Page 5, line 13-14: a concatenation (the) of vectors Equation (22): The function H_2(\pi) seems undefined. Equation (25): The inequality sign should be in the opposite direction.

Reviewer 2



In the pooled data problem there are p items, each of which is assigned 1 of d possible labels. The goal is to recover the vector of labels from n "pooled tests." In each of these tests, a "pool" (subset of the items) is chosen and we observe how many items in the pool have each label (but not which items have which). A random prior on the label vector is used: the proportion with which each label occurs in the population is fixed, and the true labelling is chosen independently among all labellings that exactly obey these proportions. This paper considers the non-adaptive case where all the pools (subsets) must be chosen in advance. The focus is mainly on the case of Bernoulli random pools: the incidence matrix of pools versus items has iid Bernoulli(q) entries for some fixed 0 < q < 1. The objective is exact recovery of the label vector, although extensions to approximate recovery are considered in the appendix. This paper proves information-theoretic lower bounds for the pooled data problem, both in the noiseless setting described above and in a very general noisy model. In the noiseless setting with Bernoulli random pools, they prove the following sharp phase transition. The regime is where the number of samples is n = c*p/log(p) for a constant c, with p tending to infinity. They find the exact constant C such that if c > C then exact recovery is possible with success probability tending to 1, and if c < C then any procedure will fail with probability tending to 1. The constant C depends on the number of labels and the proportions of each label (but not on the constant q which determines the pool sizes). The upper bound was known previously; this paper proves the tight lower bound (i.e. the impossibility result), improving upon a loose bound in prior work. The sharp information-theoretic threshold above is not known to be achieved by any efficient (i.e. polynomial time) algorithm. Prior work (citation [4] in the paper) suggests that efficient algorithms require a different threshold (in which n scales proportionally to p). The key technical insight used in this paper to prove the sharp lower bound is the following "genie argument." Let G be some subset of {1,..,d} (the label values), and imagine revealing the labels of all the items that have labels not in G. Now apply a more standard lower bound to this setting, and optimize G in order to get the strongest possible bound. The other contribution of this paper is to the noisy setting, which has not previously been studied. This paper defines a very general noise model where each observation is corrupted via an arbitrary noisy channel. They give a lower bound for exact recovery in this setting, and specialize it to certain concrete cases (e.g. Gaussian noise). The proof of the lower bound is based on a variant of the above genie argument, combined with standard lower bounds based on Fano's inequality. Unfortunately there are no existing upper bounds for the noisy setting, so it is hard to judge the strength of this lower bound. The authors do, however, argue for its efficacy in a few different ways. First, it recovers the sharp results for the noiseless case and also recovers known bounds for the "group testing" problem, which is essentially the special case of 2 labels: "defective" and "non-defective" (and you want to identify the defective items). Furthermore, the lower bound for Gaussian noise is strong enough to show an asymptotic separation between the noiseless and noisy cases: while in the noiseless case it is necessary and sufficient for n to be of order p/log(p), the noisy case requires n to be at least p*log(p). In the conclusion, the authors pose the open question of finding upper bounds for the noisy setting and provide some justification as to why this seem difficult using existing techniques. I think this is a good paper and I recommend its acceptance. The exposition is good. The sharp lower bound for the noiseless case is a very nice result, and the proof technique seems to be novel. The lower bound for the noisy case is also a nice contribution; although it is hard to judge whether or not it should be tight (since no upper bound is given), it is certainly nontrivial, using the genie argument to surpass standard methods. Specific comment: - I think there is a typo in Table 2: the heading "exact recovery" should be "noiseless." (If I understand correctly, the entire table pertains to exact recovery.)

Reviewer 3



This paper explores the pooled data problem introduced in [1], and derives a lower bound on the number of pooled tests required to identify the ground-truth labels of items. The derived bound gives two main implications: (1) The noiseless setting: Characterizing the sharp threshold on the required number of tests under i.i.d. Bern(q) measurement matrix, thereby proving the optimality of the message passing algorithm in [3]; (2) A noisy setting: Evaluating several mutual information terms that form the lower bound to investigate the effect of noise upon the performance under a Gaussian noise model. The paper is well written - the main results are clearly presented and the proofs are well streamlined together with accessible outlines. In particular, in the noiseless setting, it tightens the lower bound in [3] via a converse technique, thus showing the tightness of the upper bound in [3]. The major concerns of this reviewer are as follows: 1. (technical contribution): As mentioned in Example 2, Corollary 1 recovers the lower bounds given in the binary group testing problem [9], which can be viewed as a special case of d=2. I am wondering how distinct the noisy-case converse technique developed in this paper (which also covers the noiseless setting under the vanishing error probability criterion) relative to the one in [9], which is also based on Fano's inequality and a genie-aided argument. Perhaps a key contribution might be extending to an arbitrary d (in which a more involved combinatorics arise) and explicitly evaluating several mutual information terms being tailored for the model considered herein. If that is the case, the technical contribution looks marginal although its implication is interesting. 2. (i.i.d. measurement matrix) The above two implications are based on the i.i.d. assumption made on the measurement matrix. This assumption may be arguable though for some applications in which adaptive measurements are allowed and/or some measurement constraint (like locality constraint) is applied. In particular, compared to the adaptive measurement setting, the lower bound claimed in the noisy setting might not be that informative, as one can only do better in the adaptive setting. I am wondering how the developed bound can be extended (if possible) to a broader class of measurement models. This generalization is perhaps challenging though, due to the potential difficulty of single-letterization for beyond-iid models. 3. (Gaussian model) Not clear as to how practically-relevant is the Gaussian model under which the second implication is developed. Any important applications which respect the Gaussian model? 4. (upper bound) Developing an algorithm for the noisy setting is deferred as a future work. 5. (applications) It is not clear how the pooled data problem arises in the applications mentioned in the introduction. More elaboration may be helpful. Minor comments: 1. As for the indicator function near Eq. (2): Adding brackets in the two sets intersected can improve readability. 2. Comparison to the very recent work [6] is not clear enough. For instance, what are qmax and \Delta? What does the condition \Delta >> \sqrt{qmax} mean? More detailed explanation might help. 3. As mentioned in the paper, under the vanishing error probability criterion, the noisy-case bound recovers the one in the noiseless setting, in which the proof technique looks distinct. Any way to unify the two proofs? 4. One of the key techniques in the converse lie in the genie argument. But this part is relatively harder to follow. A more comprehensive illustration might help. -------------------- (Additional comments) I have read the rebuttal - some of the comments are addressed. Given that the converse proof in the noiseless setting is quite distinct relative to [9], the paper seems to make a technical contribution as well. Perhaps this can be further highlighted in a revision.